# Comprehensive Omics Analysis Reveals Cold-Induced Metabolic Reprogramming and Alternative Splicing in *Dendrobium officinale*

**DOI:** 10.3390/plants14030412

**Published:** 2025-01-30

**Authors:** Xinqiao Zhan, Zhangqun Li, Minxia Pang, Guoxiang Yao, Bizeng Mao

**Affiliations:** 1Institute of Biotechnology, Ministry of Agriculture Key Lab of Molecular Biology of Crop Pathogens and Insects, Zhejiang Key Laboratory of Biology and Ecological Regulation of Crop Pathogens and Insects, Zhejiang University, Hangzhou 310058, China; 2School of Pharmaceutical Sciences, Taizhou University, Taizhou 318000, China; lizhangqun@126.com; 3Zhejiang Jianjiuhe Group Co., Ltd., Ningbo 315000, China; pmx059@163.com (M.P.); yao.guoxiang@shunyun.cn (G.Y.); 4Ningbo Shunyun Electroinic Co., Ltd., Ningbo 315000, China

**Keywords:** *Dendrobium officinale*, alternative splicing, cold stress, ubiquitination, glycine metabolism

## Abstract

*Dendrobium officinale* is an economically important orchid species that is sensitive to cold stress. Understanding the molecular and metabolic mechanisms underlying its response to cold is crucial for developing strategies to improve its cold tolerance. In this study, we constructed a comprehensive cold stress response dataset for *D. officinale* and characterized its regulatory landscape in response to varying cold stress conditions. The glycine metabolism-related genes *Dca003913* and *Dca022726* play pivotal roles in both cold and drought stress adaptation, and their expression is not upregulated by hormones or fungi infection. Carbohydrate metabolism showed specific dynamic changes in freezing injury cells, which involved a variety of hormonal responses. The abundance of sphingolipids was notably higher in the freezing treatment (FT) compared to the freezing recovery (FR) plants, indicating specialized metabolic adaptations at different cold intensities. An alternative splicing (AS) analysis identified 368 DAS genes, with spliceosome pathways significantly enriched. Three key ubiquitination proteins (PKU64802, XP_020672210, and PKU75555) were found to regulate splicing factors, which showed increased abundance in cold stress. This study highlights the roles of metabolic reprogramming and RNA splicing in cold adaptation, revealing a complex molecular network activated in response to cold stress.

## 1. Introduction

Low and freezing temperatures pose significant challenges to crops, severely affecting their growth, metabolism, and geographical distribution [1,2]. Plants have evolved complex mechanisms to sense and respond to cold stress, including changes in gene expression, metabolic adjustments, and alterations in RNA processing [3,4]. Multi-omics has been used to study a series of physiological and biochemical changes in plants. For example, transcriptomic and lipidomic analyses found that 59 cold-tolerant genes were involved in lipid metabolism, highlighting the protective roles of increased digalactosyldiacylglycerol (DGDG) and sulfoquinovosyldiacylglycerol (SQDG) levels against photosynthetic damage and upregulated phosphatidate-phosphatase (PAP1) and phosphatidate-cytidylyltransferase (CDS1) in preventing membrane lipid peroxidation, contributing to a comprehensive metabolic model for peanut cold tolerance [5]. Cold-acclimated seedlings of *Sonneratia apetala* exhibit enhanced cold tolerance through physiological adjustments and differential accumulation of proteins involved in ROS scavenging, photosynthesis, energy metabolism, carbohydrate metabolism, cofactor biosynthesis, and protein folding, with key regulators including translation elongation factor G, chlorophyll A-B binding protein, and ascorbate peroxidase [6]. In maize, decreasing the temperature from 20 to 16 °C alters 50% of metabolites and 18% of proteins, with specific examples including decreasing levels of photosynthesis-related proteins and increasing levels of metabolites such as trans-acetate and hydroxycinnamate derivatives, which could serve as potential markers for breeding cold-tolerant varieties [6]. Analyses of the integrated transcriptome–metabolome identified 3140 differentially expressed genes, including those involved in jasmonic acid synthesis and signaling, and found increased levels of branched-chain amino acids such as leucine, isoleucine, and valine, which together contribute to the seed browning and chilling response in *Capsicum annuum* fruit [7]. The expression of transcription factors such as C-repeat binding factors (CBFs) and the subsequent induction of downstream cold-responsive genes are well-documented [8]. However, little is known about the complex interplay between transcriptional, translational, and metabolic changes in response to cold stress.

Alternative splicing (AS) enables plants to produce diverse transcripts from a single gene, significantly enhancing the functional diversity of the transcriptome [9]. AS plays a critical role in plant responses to various environmental stresses, including cold stress [9]. The cold stress responses in plants are regulated by several transcription factors (TFs), such as the bZIP (basic leucine zipper protein), MYB (*v-myb* avian myeloblastosis viral oncogene homolog), and WRKY (conserved amino acids) families, which are associated with regulating AS to produce differences in protein function [10,11]. By generating multiple protein isoforms or non-coding RNAs, AS contributes to the complexity of the plant’s response to changing environmental conditions. Thus, identification of AS events is important in transcriptional control. Currently, RNA-Seq based on next-generation sequencing technology is extensively employed in transcriptomics research in plants. It is a high-throughput method enabling the precise quantification and annotation of AS events. Studies on model plants have provided evidence that RNA-Seq technology discloses AS in 61% and 48% of multi-exon genes, in [12,13], respectively. However, the regulation of AS at the post-transcriptional level in response to both biotic and abiotic stresses still lacks study.

*D. officinale* is a valuable orchid species, which is mainly distributed in southern China [14,15]. *D. officinale* contains numerous bioactive extracts such as polysaccharides, alkaloids, phenols, terpenes, coumarins, and flavonoids [16,17]. Wild *D. officinale* grows on semi-shady rocks in mountainous areas at an altitude of 800–1600 m, and its quantity is scarce. At present, it has been widely cultivated artificially in southern China. The cultivars we studied were located in Zhejiang Province. In winter, the temperature can drop to around −10 °C. Although the cultivated environment is more stable and milder, extreme low temperatures in winter still cause a decrease in *D. officinale* production. At present, only two studies have reported on cold stress in *D. officinale* [18,19]. The cold stress responses of *D. officinale* are still unclear, which restricts the potential of gene engineering to attain enhanced cold tolerance. Moreover, the basic research into *D. officinale* still focuses on gene classification analysis [20,21]. The specific effects of cold stress on AS events and the regulation of splicing factors in *D. officinale* remain largely unexplored. The objective of this study is to investigate the molecular and metabolic responses of *D*. *officinale* to cold stress, with a focus on the dynamics of alternative splicing and the regulation of splicing factors in the treatment of cold stress. Understanding these processes will provide insights into the mechanisms underlying cold adaptation and offer potential targets for improving cold tolerance in *Dendrobium*.

## 2. Results

### 2.1. The Core Pathway Involved in the Cold Response of D. officinale

To study the core response pathway of cold stress in *D*. *officinale*, we integrated two transcriptional datasets of cold treatment. The sample names for the datasets were CK1 (25 °C), CK2 (20 °C), CA (8 °C), FT1 (0 °C), FT2 (−6 °C), and FR (8 °C, recovery). A principal component analysis (PCA) indicated partial overlap between the CK and cold treatments (Figure 1a). The transcriptional data were processed through a Bayesian framework to eliminate batch effects. A PCA and hierarchical cluster analysis showed that the CK and cold treatment were obviously separated into two groups (Figure 1b,c). Furthermore, Gene Set Enrichment Analysis (GSEA) revealed the downregulation of eight core metabolic pathways during cold treatment, including nucleotide, amino acid, lipid, and secondary metabolism (Figure 1d). C5-branched dibasic acid metabolism and the glycine, serine and threonine metabolism related pathway levels were enhanced during extensive cold treatment compared to the CK group (Figure 1d). Six representative pathways were selected from the GSEA. A hierarchical cluster analysis showed these pathways related to gene levels were statistically separated into two clades based on treatment and were basically consistent under different cold treatments (Figure 1e). Random forest methods were used to identify the feature genes associated within the cold and CK groups. The top 10 genes with high accuracy were identified, among which 4 genes were abundant in cold, and 6 genes were abundant in the CK group (Figure 1f). Cold treatment induced cell dehydration and rupture, due to the formation of ice crystals in the extracellular space of the cell membrane. Thus, we detected the expression levels of three genes in leaves during the drying treatment and rewetting phase. *Dca003913* and *Dca022726* were induced by the dry treatment and were restored in expression during the rewetting phase (Figure 1g). In addition, the content of glycine increased during the dry treatment and recovered during rewetting (Appendix A). However, the expressions of *Dca003913* and *Dca022726* were not increased by hormones or fungus infection (Figure 1h). These results suggested that the gene induction of glycine degradation was a key component in cell dehydration.

The differentially expressed genes (DEGs) were identified using a fold change (FC) of >2 and an adjusted *p* value of <0.05 as the cutoffs. In total, 2679 genes were upregulated and 2572 were downregulated in the cold vs. CK groups (Appendix A). The top 20 genes were labeled, and they were involved in multiple pathways, including secondary metabolism, carbohydrate metabolism, and stress response (Appendix A). These genes were not only clustered under cold treatment but also varied according to their temperature changes. Thus, *D*. *officinale* might have a wide response mechanism to cold and a temperature-specific response mechanism.

### 2.2. Temperature-Specific Response in D. officinale

Transcriptome analysis suggested that *D. officinale* exhibits temperature-specific gene expression responses. To functionally characterize specific genes in cold treatment, we performed a WGCNA (Weighted Correlation Network Analysis) with transcriptomes and samples. This yielded seven distinct co-expression modules, each assigned a unique color and correlated with six treatments (Figure 2a). Four of the modules were clearly temperature-dependent, such as FT1 and FT2, exhibiting similar modules and significantly correlated (*p* < 0.05) with blue and yellow (Figure 2a). We found that the essential relationship between the modules FT2 and FR was most apparent through clustering analyses (Figure 2b). Furthermore, we merged 5251 DEGs of the cold vs. CK treatments with genes with the yellow and blue modules, and 487 overlapping genes were selected to analyze the KEGG (Kyoto Encyclopedia of Genes and Genomes) enrichment pathways (Figure 2c). KEGG terms related to carbon metabolism and photosynthesis were significantly enriched (*p* < 0.05) during freezing treatment (Figure 2d). Freezing conditions caused serious damage to the cell membrane fluidity. However, lipid metabolism did not emerge as a significant pathway in this gene set. To further analyze this consistently impacted set, we compared the carbon metabolism related gene levels in the CK1, FT2, and FR treatments. Examples include transcripts for the fructose-bisphosphate aldolase, malate dehydrogenase, and ribulose bisphosphate carboxylase, all of which were more abundant in the FT2 and FR treatments compared to the CK1 treatment (Figure 2e). The metabolic processes involved in these genes were not only involved in carbon metabolism but were also related to plastids and chloroplasts. We further identified the common genes associated with the FR-related modules and 5251 DEGs. In total, 261 overlapping genes were significantly enriched in five pathways (*p* < 0.05), including pathogen interaction, signal transduction, and sugar metabolism (Figure 2f). The five genes related to auxin, cytokinin, gibberellin, and brassinosteroid signal transduction were significantly upregulated in the FT2 and FR treatments compared to the CK1 treatment (Figure 2h). Seven genes involved in sugar metabolism had consistently increased mRNA levels in the FR treatment compared to the FT2 and CK1 treatments, including glycosyltransferase, pyruvate kinase, and trehalose-6-phosphate phosphatase (Figure 2i). These results suggested that the carbohydrate metabolism showed specific dynamic changes in freezing injury cells, which involved a variety of hormonal responses.

### 2.3. Metabolite-Specific Accumulation Under Cold Treatment in D. officinale

In order to identify the core metabolites specifically accumulated under cold stress, metabolomic profiling was performed. The distribution of metabolites across various treatments, based on their proportion, revealed that 60% of lipids, amino acids, and their secondary derivatives accumulations were closely associated with the FR treatment (Figure 3a). The details of these metabolites are shown in Appendix A. The top 10 metabolites with respect to the mean intensity across the three treatments were selected. In comparison to the CK and FT treatments, the levels of four metabolites, namely oleamide, apigenin glucoside, rutin, and vicenin, exhibited a distinct increase in the FR treatment (Figure 3b). Under cold conditions, the unsaturated glycolipid monogalactosyldiacylglycerol (MGDG, 36:5) demonstrated an elevation relative to the CK group (Figure 3b).

To comprehensively explore the lipid profile alterations under cold stress, 5981 lipids were classified into seven categories (Appendix A). Most lipids were concentrated around certain regions and increased in the FT and FR treatments (Figure 3c). We further analyzed the specifically accumulated lipids in the FT and FR treatments (Appendix A and Figure 3d,e). The species numbers of sphingolipids were 3.4-fold more extensive in the FT group compared to the FR treatment. Among them, ceramides (Cer) were the most dramatically changed sphingolipid. Glycerophospholipid and saccharolipid species were decreased 3.4-fold and 1.5-fold in the FT treatment compared to the FR treatment, respectively (Figure 3d,e). The ratio of MGDG and DGDG decreased two-fold in the FR treatment compared to the FT treatment. Another kind of saccharolipid, SQDG, was particularly increased in the FR treatment compared to the FT treatment (Figure 3d,e). These results suggest that sphingolipid and glycerophospholipid metabolism are related to enhance cold tolerance.

Transcriptome integration analysis identified several amino acid metabolites involved in cold treatment. However, further quantification and comprehensive analysis of the protein expression were important in revealing the metabolic processes during cold treatment (Appendix A). Most protein positions were left of center and were close to the FT and CK groups (Figure 3f). The signal transduction and secondary metabolite biosynthesis proteins were very intensive, with a concomitant increase in the FR compartment (Figure 3f). To further analyze the consistent impact of transcription and translation during cold, we compared the log_2_ fold change (FC) values between the cold and CK samples in the mRNA and protein levels. Of the 1671 protein values, 41% (682 proteins) showed at least two-fold differences in mRNA and/or protein levels in the FT vs. CK groups (Figure 3g). In the FR vs. CK groups, 768 proteins showed at least two-fold differences in mRNA and/or protein levels (Figure 3h). The mauve points (n6 sector) showed significant increases only in protein abundance and not for mRNA. The protein number of the FR treatment vs. the CK group (184 proteins) was higher than that of the FT group vs. the CK group (14 proteins). These results implied that the rise in most proteins during the FR treatment was independent of transcription and/or translation. For the gene enrichment analysis of the FT group and FR treatment in the n2 sector (indigo points), n3 sector (orange points), and n6 sector (mauve points), numerous biosynthesis processes were identified whose increased mRNA and/or protein levels revealed obvious differences between the FT group and FR treatment (Figure 3i,j). For the n2 sector, the FT treatments induced the mRNA levels of pentose metabolism and fatty acid biosynthesis, and the FR treatment induced the mRNA levels of nucleotide sugar metabolism and amino acid biosynthesis. For the n3 sector, linoleic acid metabolism was especially enriched (during the FT treatments) for both protein and mRNA. For the n6 sector, amino acid biosynthesis, including glutathione metabolism, was only significantly enriched during the FR treatment at protein levels. The protein number related to thylakoids in the FT group was lower than that in the FR treatment, which was probably driven by catabolic routes in the presence of autophagy.

### 2.4. Identification of Key Splicing Factors in Response to Cold in D. officinale

Notably, the spliceosome pathway was significantly enriched in the individual sector during cold treatment (Figure 3i,j), which is probably driven by pre-mRNA processes. Five main types of AS events were studied in two *D*. *officinale* genome annotation versions in NCBI (GCA_001605985.2 and GCF_001605985.2). However, 12,569 AS events were found in the genome (GCF_001605985.2, Figure 4a). Among, them, A3SS (alternative 3′ splice sites) was the most abundant type (41%), followed by EX (exon skipping, 24%), A5SS (alternative 5′ splice sites, 22%), IR (intron retention, 11%), and MX (exclusive exons, 2%). We further identified the AS events (PSI > 0) in six temperature treatments. Low temperature exposure triggered a rapid AS response with a significantly higher number of AS events. In contrast, the FT and FR treatments significantly decreased the number of AS events compared to the CK1/2 treatment (Figure 4b). We merged the AS events for the normal and cold conditions. We discovered that the IR and MX of the cold group were the same as that of the CK group. The CA treatment induced an increase in A3SS, A5SS, and EX compared to the CK group, and all three decreased in the FT group and FR treatment. However, A3SS and EX had an increasing tendency in the FR treatment (Figure 4c). This suggests that *D*. *officinale* tends to induce A3SS, A5SS, and EX under low temperatures and sub-zero temperatures that seriously affect RNA splicing.

To characterize the dynamics of AS during exposure to cold treatment, we compared the differences in AS events at different temperature conditions. The results revealed that increased differential alternative splicing (DAS) events occurred in the cold compared to the controls. FT-induced AS events persisted throughout the FR period (Figure 4d). Furthermore, we merged 368 DAS genes from six comparisons that were clustered across five regions and exhibited substantial variation in the expression levels (Figure 4e). Functional annotation of the DAS genes revealed that the spliceosome was specially enriched during the CA and FR treatments (Figure 4f). Moreover, carbon metabolism and amino acid biosynthesis were specifically enriched in the FT group and FR treatment compared to the CA treatment (Figure 4f). Most of the transcripts of the DAS genes, including spliceosome, carbon, and amino acid metabolism, had higher expression in the CA treatment compared to the FT group and FR treatment (Appendix A). We selected three DAS genes to examine their transcript expression. The *malate dehydrogenase* (*LOC110113907*) and *cysteine synthase* (*LOC110093223*) transcript expression levels were especially induced in the CA treatment. The *Zinc knuckle* transcript (*LOC110095000*) expression level was suppressed by cold treatment (Appendix A). This result confirmed the accuracy of the DAS gene identification in the transcriptome.

To verify the key splicing factors in response to cold, we focused on the expression patterns of the 368 DAS genes. They were grouped into two classes according to expression levels, and three DAS genes were identified in class 1 (Figure 4g). Among them, *E2 ubiquitin-conjugating enzyme* was specially induced during cold treatment and *ATP binding actin* decreased during cold treatment (Figure 4h and Appendix A). Ubiquitination is an important pathway of protein modification. We speculated that ubiquitin proteins mediate the regulation of splicing factors during cold treatment. We further analyzed the abundance levels of 38 ubiquitin proteins and the expression of 324 splicing factors under cold treatment and found that the splicing factor transcript levels were strongly regulated by cold treatment. However, the expression of the vast majority of splicing factor genes was down-regulated upon exposure to cold treatment (Appendix A). Spearman correlation analysis showed seven splicing factor genes were positively correlated with 13 ubiquitin proteins (Figure 4i). Furthermore, most ubiquitin proteins increased their abundance levels in the CA treatment and FT group compared to the CK group. Most of the splicing factor genes were also significantly decreased in the CA and FR treatments compared to the CK group (Figure 4j,k). Three splicing factor genes were retrieved from the proteome in response to the FT and FR treatments, including *Dca015217*, *Dca012567*, and *Dca027386*, which respectively encoded an RNA binding motif, an RNA recognition motif, and a serine/arginine-rich protein. All of their abundance levels significantly increased in the FT group and FR treatment compared to the CK group and decreased in the CA treatment compared to the CK group (Figure 4l). Furthermore, ten ubiquitin-proteins-related genes were expressed in all tissues, and the expression levels of *PKU64802*, *XP_020672210*, and *PKU75555* were highest in leaves (Appendix A). *PKU64802* encoded the ubiquitin carboxyl-terminal hydrolase that directly involved the intracellular protein homeostasis and signaling. *XP_020672210* and *PKU75555* belong to the E2 ubiquitin-conjugating enzyme. To verify the function of these genes in the cold response in *D*. *officinale*, as-ODNs were injected into mature leaves for 24 h during FT treatment. We found that these genes’ expression levels were decreased by as-ODNs during FT treatment (Appendix A). The inhibition of *PKU64802* expression resulted in a significant increase in *Dca015217* and *Dca027386* expression during FT treatment (Appendix A). Thus, these splicing factors could be the key genes that were fine-regulated by ubiquitin systems in response to different temperature treatments, suggesting a unique role for AS in cold adaptation.

## 3. Discussion

This study addresses gaps in understanding the cold stress response in *D. officinale* through a multi-omics analysis of transcriptional and AS regulation. As proof-of-principle to evaluate the value of our dataset, we employ it to reveal the complex interplay between temperature, transcriptomic changes, alternative splicing dynamics, splicing factor regulation, and metabolic adjustments in *D. officinale* under cold stress (Figure 5).

Temperature extremes are major stressors for staple crops such as rice, spinach, and maize. Under cold stress, crops exhibit extensive differences in metabolic accumulation. A metabolomics study on rice revealed that cold stress induced the expression of *OsHPL1*, a CYP74 family member, and led to the accumulation of 12-oxo-phytodienoic acid and jasmonates [22]. Following cold stress, maize plants exhibit specific metabolomic changes, including alterations in biomass allocation, shikimate and its aromatic amino acid derivatives, and other non-polar metabolites [23]. Metabolome changes in spinach leaves were analyzed after short and long term freezing at −4.5 °C. Nineteen selected metabolites (e.g., aromatic amino acids, lactic acid) were grouped into signaling, injury, and recovery clusters, with diverse accumulations reflecting different cold stress responses and mechanisms [24]. However, the metabolic responsiveness of *D*. *officinale* under cold stress is still unclear. A thorough analysis of the responses of *D*. *officinale* to cold stress uncovers a multi-dimensional adaptive strategy encompassing both transcriptomic and metabolic modifications. At the transcriptomic scale, there is a distinct demarcation between the control and cold treatments in the plant. Key metabolic pathways, including nucleotide, amino acid, lipid, and secondary metabolism, experience downregulation (Figure 1d). This downregulation presumably indicates a reallocation of energy resources to handle the stress. In contrast, the upregulation of the C5-branched dibasic acid metabolism and the glycine, serine, and threonine metabolism pathways implies a specific adaptation mechanism for maintaining cellular homeostasis under cold stress (Figure 1e). Glycine is one of the basic amino acids that make up proteins. It plays a role in plant stress resistance, including helping plants cope with nitrogen, drought, cold and other adverse environments [25,26,27]. The identification of key genes involved in glycine metabolism, such as *Dca003913* and *Dca022726* (Figure 1f–h), further supports the hypothesis that glycine metabolism plays a critical role in the plant response to dehydration caused by cold stress.

The temperature-specific response of *D*. *officinale* to cold stress is characterized by distinct transcriptomic and metabolic signatures. WGCNA highlights the presence of temperature-dependent co-expression modules, indicating that the plant adapts to cold stress through the activation of specific genetic programs (Figure 2). The enrichment of carbon metabolism and photosynthesis pathways in freezing conditions suggests that *D*. *officinale* reconfigures its metabolic processes to sustain basic functions under extreme cold (Figure 2d,g). During cold stress, plants adjust their osmotic potential by accumulating solutes such as soluble sugars within their cells [28]. Soluble sugars are generally regarded as an antioxidant to protect against anoxic injury [29]. A proteomic analysis of ABA-deficit barley and a wild type showed that ABA deficiency affected proline, soluble protein, and H_2_O_2_, as well as modifications in starch and sucrose biosynthesis, influencing low temperature tolerance [30]. The observed changes in carbohydrate metabolism and hormonal responses, particularly in freezing injury cells, indicate a complex interplay between metabolic and signaling pathways [31]. The plasma membrane (PM) senses cold stress and produces signaling lipids via enzymes such as phospholipases. Signaling lipids (e.g., phosphatidic acid, phosphoinositides, and sphingolipids), which account for <1% of total lipids, increase under stresses [32]. Sphingolipids in the PM, which are involved in lipid raft formation and composition within microdomains, help plants detect specific temperature ranges. [33]. Cold stress triggers a transient Ca^2+^ influx into the cytosol via PM rigidification-activated mechanosensitive or ligand-activated Ca^2+^ channels [34]. However, the direct link of sphingolipids and calcium channels is still unknown. We speculate that sphingolipids and calcium channels play crucial and complementary roles in the plant’s initial perception of cold stress.

The dynamics of AS events and the regulation of splicing factors in *D*. *officinale* under cold stress provide novel insights into the molecular mechanisms underlying cold adaptation. AS, generating alternative 5′ and/or 3′ splice sites, is significant for plants’ abiotic stress response and adaptation [35]. AS events increased during mild cold stress but decreased under freezing conditions, indicating a fine-tuned regulatory mechanism optimizing gene expression (Figure 4b). The enrichment of spliceosome-related genes and the identification of specific splicing factors that are regulated by the ubiquitination pathway highlight the role of post-translational modifications in controlling splicing events (Figure 4i). Serine/arginine-rich proteins are involved in the composition and AS of pre-mRNA in plants, and they are crucial for regulating transcriptome and proteome diversity [36]. Overexpression of the serine/arginine-rich protein SR45 reduces the sensitivity of plants to abscisic acid during the early developmental stage [37]. The observed changes in the expression and abundance of splicing factors, such as serine/arginine-rich protein *Dca012567*, indicate a complex regulatory network that is responsive to temperature changes (Figure 4l). Ubiquitination regulation is involved in numerous cellular processes. It marks proteins for degradation, thereby controlling the levels of specific proteins within the cell [38]. However, a role for the post-translational modification (PTM) of AS components has emerged in recent years, such as the ubiquitin/26S proteasome pathway in plants [39]. PTM is associated with phosphorylation, acetylation, and more and adjusts protein activity and subcellular localization in every cellular process [40]. Among them, ubiquitination or ubiquitin-like modification is a key regulator in eukaryotic cells [41]. These modify target proteins via conjugation with small molecules, causing changes in their degradation or new interactions. Many proteins, including spliceosome-associated ones, undergo such modifications, and PTMs of spliceosome components are major regulatory factors in yeast and human cells [42,43]. The expression patterns of DAS genes provide insights into the regulation of splicing factors in response to cold stress (Figure 4). Among these, E2 ubiquitin-conjugating enzyme is specifically induced during cold treatment, while zinc knuckle factor expression decreases (Figure 4h). The abundance of ubiquitin proteins increases in the CA treatment and FT group compared to the CK group, while the expression of the majority of splicing factor genes decreases in the CA and FR treatments compared to the CK group (Figure 4j–l). These findings suggest that the regulation of splicing factors by the ubiquitin system plays a critical role in the response to different temperature treatments, affecting alternative splicing dynamics and contributing to cold adaptation in *D*. *officinale*.

The multi-omics analysis of *D*. *officinale* under cold stress provides a comprehensive view of the molecular and metabolic adaptations that enable the plant to cope with cold stress (Figure 5). The findings contribute to our understanding of the plant’s temperature-specific response and offer potential targets for improving cold tolerance in this species.

## 4. Materials and Methods

### 4.1. Plant Material and Treatments

Two-year-old cultivated *D*. *officinale* plants were grown in soil under controlled greenhouse conditions (25 °C/23 °C day/night, 75% humidity, 12-h photoperiod, 80 μmol photons m^−2^ s^−1^) at Zhejiang University (Hangzhou, China). Cold treatment conditions were performed according to the previous study [18,19]. In brief, plants were subjected to different temperature treatments, including control (CK1/2): maintained at 25 °C and 20 °C; cold acclimation (CA): exposed to 8 °C for 24 h; freezing treatment (FT): exposed to −6 °C for 3 h; and freezing recovery (FR): after FT, plants were allowed to recover at 8 °C for 24 h. The leaf samples were carefully collected from the above treated plants. To ensure the integrity of the samples and prevent any potential degradation of biological components, all the freshly collected leaf samples were promptly immersed in liquid nitrogen.

### 4.2. Multi-Omics Analysis

Transcriptome, proteome, lipidome and, metabolome data were obtained from *D*. *officinale* leaf samples subjected to the different temperature treatments, as described above. The methodology of the transcriptome data analysis was performed according to previous studies [44]. In brief, total RNA was extracted from leaf samples using TRIzol reagent (Thermo Scientific, Waltham, MA, USA), according to the manufacturer’s instructions. Poly-A containing mRNA molecules are selectively purified from total RNA using oligo(dT) attached to magnetic beads (Invitrogen, Waltham, MA, USA). These mRNA molecules undergo reverse transcription to generate cDNA strands, which are then converted into a sequencing library through adapter ligation and PCR amplification, ensuring quality and quantity via the mRNASeq sample preparation kit (Illumina, San Diego, CA, USA), before being sequenced on a high-throughput platform to produce millions of short reads. Raw reads were filtered to remove adaptor sequences and low-quality reads. Clean reads were aligned to the *D*. *officinale* reference genome [15]. Differential gene expression analysis was conducted using DESeq2 (version 1.30.1) [45] with a threshold of |log_2_(fold change)| ≥ 1 and adjusted *p*-value < 0.05. The transcriptional data were removed for batch effects and other unwanted sources of variation based on an empirical Bayesian framework [46].

For proteome analysis, protein extraction was performed using RIPA buffer (Thermo Scientific, USA) containing protease inhibitors (Roche., Rotkreuz, Switzerland). Protein concentration was determined using the Bradford assay (Bio-Rad, Hercules, CA, USA). Proteins were digested to peptides using trypsin (Promega, Madison, WI, USA) overnight at 37 °C. Peptides were separated via UHPLC (Bruker, Bremen, Germany) on a C18-RP column using a gradient from 2% to 80% ACN with 0.1% formic acid at 0.3 μL/min and 50 °C. The LC system was coupled to a timsTOF Pro2 mass spectrometer (Bruker, Germany) equipped with a CaptiveSpray ion source, operating under 1400 V capillary voltage and scanning *m*/*z* from 100 to 1700. Ion mobility separation was performed within the range of 0.7 to 1.4 Vs cm^−2^, with accumulation and ramp times set to 100 ms for near 100% duty cycle. Raw data were processed using FragPipe (v17.1) for peptide and protein identification, and *Dendrobium* taxid (37818) of NCBI was performed to annotate. Differential protein expression analysis was conducted using a Wilcoxon Rank-Sum Test with a threshold of |log_2_FC| ≥ 1 and adjusted *p*-value < 0.05.

For lipidome analysis, leaf samples were homogenized in a mixture of methanol:chloroform:water (2:1:0.8, *v*/*v*/*v*) and vortexed. After centrifugation, the lower organic phase was collected, dried with nitrogen gas, and resuspended in chloroform:methanol (2:1, *v*/*v*). A 3 μL aliquot was injected into UHPLC (SHIMADZU, Kyoto, Japan) equipped with a CSH C18 column for separation using a gradient of ACN/isopropanol (1:9, *v*/*v*) with 0.1% formic acid and 0.1 mM ammonium formate at 300 μL/min. The eluent was analyzed by Q-Exactive Plus mass spectrometer (Thermo Scientific, Walham, MA, USA) with ESI parameters set to 300 °C source temperature, 350 °C capillary temperature, and 3000 V ion spray voltage, scanning *m*/*z* from 200 to 1800 with a 50% S-Lens RF level. Raw data were processed using LipidSearch (version 4.1, Thermo Fisher Scientific, Waltham, MA, USA) for lipid identification and quantification. Differential lipid analysis was conducted using PLS-DA with a threshold of VIP > 1 and *p*-value < 0.05.

For metabolome analysis, samples were thawed on ice and metabolites extracted by adding precooled 50% methanol, vortexing for 1 min, and incubating at 4 °C for overnight. UHPLC (Thermo Scientific, USA) was used to separate metabolites with a gradient elution of water and acetonitrile, both containing 0.1% formic acid, at a flow rate of 0.4 mL/min, and the injection volume was 4 µL. These separated products were detected using an Orbitrap Exploris 240 Mass Spectrometer (Thermo Scientific, USA) operated in positive and negative ion modes, with curtain gas set to 30 PSI, ion source gases at 60 PSI, and ion source temperature at 650 °C; ionspray voltages were 5000 V and 4500 V for positive and negative modes, respectively. Metabolites were annotated by matching *m*/*z* values to the KEGG database within a 10 ppm mass accuracy, and validated using isotopic distribution and an in-house fragment spectrum library. Differential metabolite analysis was conducted using PLS-DA with a threshold of VIP > 1 and *p*-value < 0.05.

### 4.3. Alternative Splicing Analysis

Alternative splicing (AS) events were identified from the RNA-seq data using SUPPA2 (version 1.2) [47]. Five main types of AS events were considered: alternative 3′ splice sites (A3SS), alternative 5′ splice sites (A5SS), exon skipping (EX), intron retention (IR), and mutually exclusive exons (MX). The percent spliced-in (PSI) values were calculated for each AS event, and differential AS (DAS) events were identified using a cutoff of |ΔPSI| > 0.1 and FDR < 0.05. The AS events were further analyzed to understand the dynamics of splicing under cold treatment, and the DAS genes were annotated using clusterProfiler for functional enrichment analysis [48].

### 4.4. Real-Time Quantitative PCR

Total RNA was isolated from different tissues using the TransZol reagent (TransGen Biotech, Beijing, China). RNA extracts were treated with DNaseI (NEB, Hitchin, UK) to eliminate DNA contamination. First-strand cDNA was produced from the RNA template by reverse transcription using the TIANscriptRTKit according to the manufacturer’s instructions (TransGen Biotech, Beijing). Real-time quantitative PCR processes were performed according to the previous study [49]. In brief, processes was conducted with the subsequent cycling conditions: an initial denaturation at 95 °C for 1 min; followed by 40 cycles, each consisting of denaturation at 95 °C for 30 s, annealing at 55 °C for 30 s, and extension at 72 °C for 30 s; finally, a terminal extension step at 72 °C for 10 min to complete DNA synthesis. The primers used are listed in Appendix A.

### 4.5. Gene Suppression Using Candidate Antisense Oligonucleotides

For gene suppression in *D. officinale* leaves, the operation was performed according to previous study [50]. In brief, candidate antisense oligonucleotides (asODN) that are complementary to a segment of the target gene were chosen via Soligo online software (version 2.2, Appendix A). A total of 0.2 mL of asODN solution (20 μM) was injected into the mature leaves during in the CK and FT treatments, and leaves injected with sODN were used as control. After 24 h, leaves were harvested for real-time quantitative PCR.

### 4.6. Statistical Analysis

Statistical analyses were performed using R (version 4.0.2), and an ANOVA was applied to compare the differences between the two groups. Data were also treated by hierarchical clustering using the R package pheatmap and by PCA using the R package FactoMineR. Correlation analysis between splicing factors and ubiquitin proteins was performed using the Spearman rank correlation coefficient.

## Figures and Tables

**Figure 1 plants-14-00412-f001:**
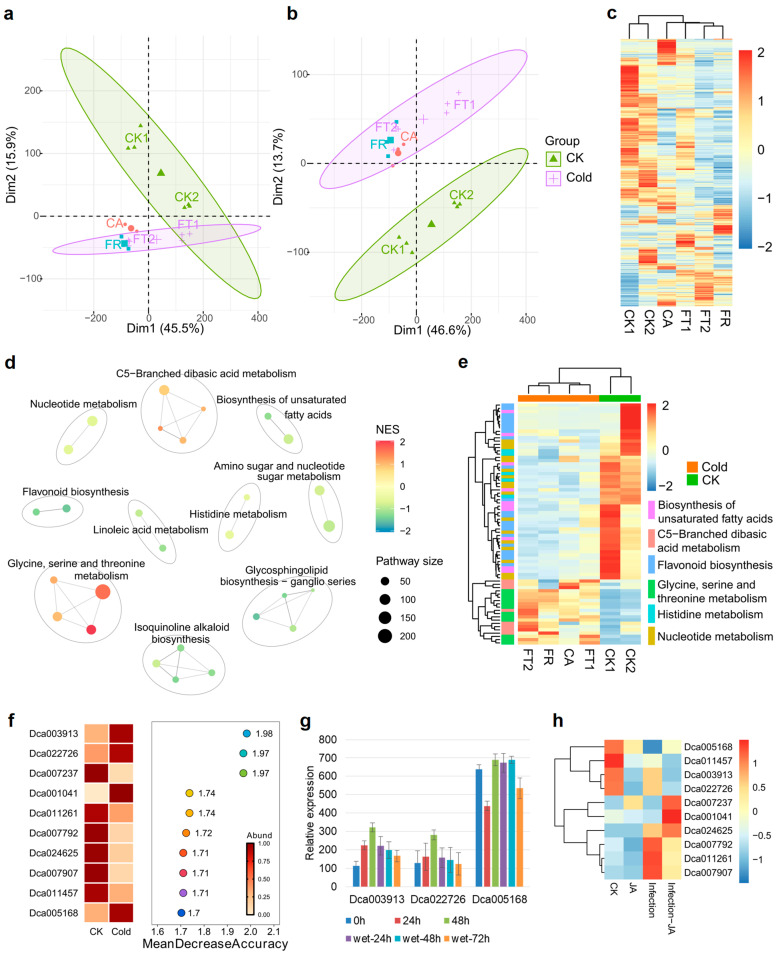
Glycine metabolism as a core pathway involved in cold stress. (**a**) PCA plot showing the separation between control conditions (CK1 and CK2) and cold treatments (FT1 and FT2, along with other cold treatments). (**b**) PCA plot after removing batch effects and other unwanted sources of variation. The CK and cold treatments are clearly separated. (**c**) Hierarchical clustering heatmap of transcriptional data showing the grouping of the CK and cold treatments. (**d**) GSEA results indicating pathways that are down- or up-regulated in the cold treatments compared to CK treatments. (**e**) Hierarchical clustering heatmap of selected pathways showing the expression patterns of genes related to these pathways across different treatments. (**f**) Random Forest analysis identifying the top 10 genes with a high accuracy in distinguishing between cold and CK conditions. Expression levels of genes abundant in cold versus CK conditions are indicated. (**g**) Expression profiles of selected genes during dry treatment and rewetting phases. Values are means ± S.D. (*n* = 3). (**h**) Expression profiles of selected genes in response to hormonal JA and fungal infection treatments.

**Figure 2 plants-14-00412-f002:**
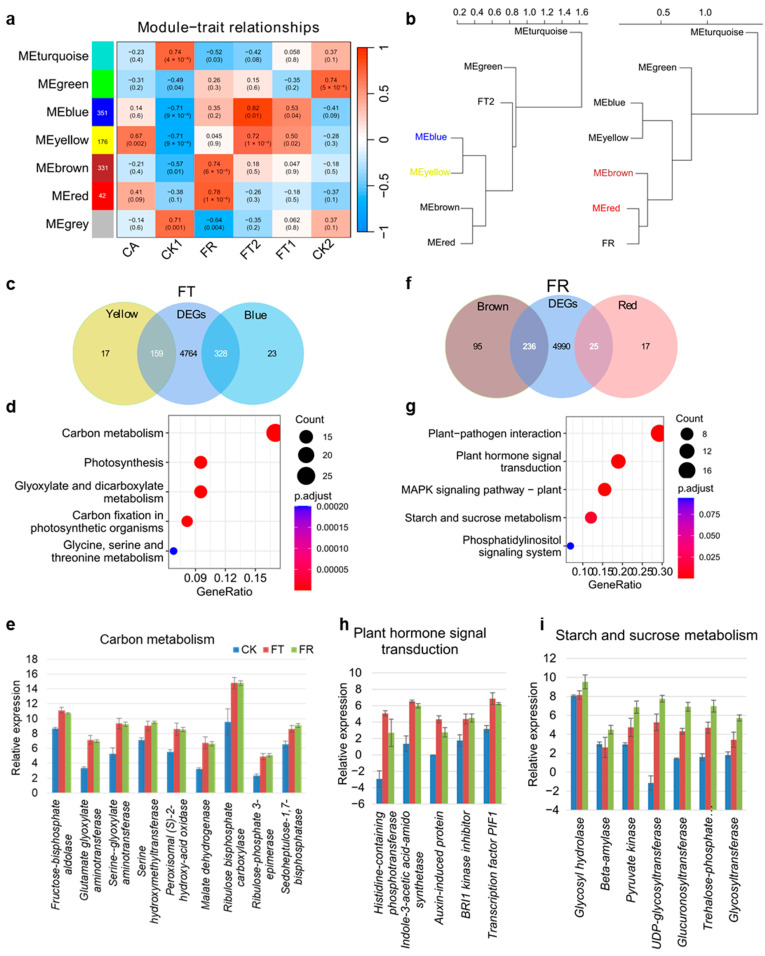
Temperature-specific response in *D. officinale*. (**a**) WGCNA showing seven distinct co-expression modules correlated with different treatments. (**b**) Clustering analysis highlighting the relationship between modules and specific treatments, particularly the FT2 and FR treatments. (**c**) Venn diagram showing the overlap between 5251 differentially expressed genes (DEGs) and genes within the yellow and blue modules. (**d**) KEGG pathway enrichment analysis of the overlapping genes from (**c**), indicating pathways significantly enriched during freezing treatments. (**e**) The expression levels of genes involved in carbon metabolism-related processes across the CK, FT, and FR treatments. Values are means ± S.D. (*n* = 3). (**f**) Venn diagram showing the overlap between genes from FR-related modules and 5251 DEGs. (**g**) KEGG pathway enrichment analysis of the overlapping genes from (**f**), indicating pathways significantly enriched during recovering treatments. (**h**) Hormone signaling-related genes that are significantly up-regulated in the FT and FR treatments compared to CK treatment. Values are means ± S.D. (*n* = 3). (**i**) Sugar metabolism-related genes showing consistent increases in mRNA levels in the FR treatment compared to the FT and CK treatments. Values are means ± S.D. (*n* = 3).

**Figure 3 plants-14-00412-f003:**
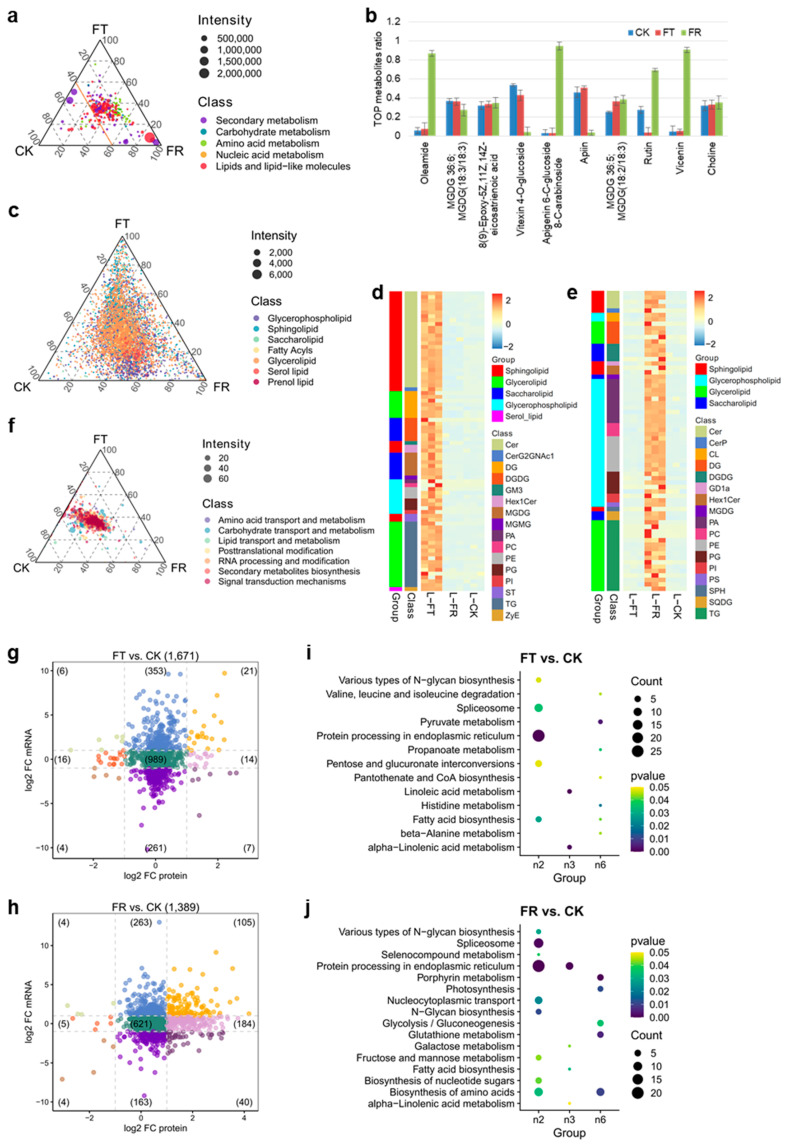
Metabolite-specific accumulation under cold treatment in *D. officinale*. (**a**) Distribution of metabolites according to their proportion in different treatments, highlighting the abundance of lipids, amino acids, and secondary derivatives in the FR treatment. (**b**) The top 10 metabolites with the highest intensity means across treatments. Specific metabolites that are increased in the FR treatment are highlighted. Values are means ± S.D. (*n* = 6). (**c**) The lipids were classified into seven categories, illustrating the progressive increase in the FT group and FR treatment. (**d**) The top proportion of metabolites in the FT group compared to the CK group and FR treatment. (**e**) The top proportion of metabolites in the FR treatment compared to CK and FT groups. (**f**) Scatter plot showing the distribution of protein positions across treatments, with an emphasis on signal transduction and secondary metabolite biosynthesis proteins. (**g**) Scatter plot comparing log_2_ fold changes in mRNA and protein levels for the FT vs. CK groups. (**h**) Scatter plot comparing log_2_ fold changes in mRNA and protein levels for the FR vs. CK groups. (**i**) Gene enrichment analysis for the n2 sector (indigo points) showing pathways with increased mRNA levels during the FT and FR treatments. (**j**) Gene enrichment analysis for the n3 sector (orange points) and n6 sector (mauve points) showing pathways with increased protein levels, particularly amino acid biosynthesis during the FR treatment.

**Figure 4 plants-14-00412-f004:**
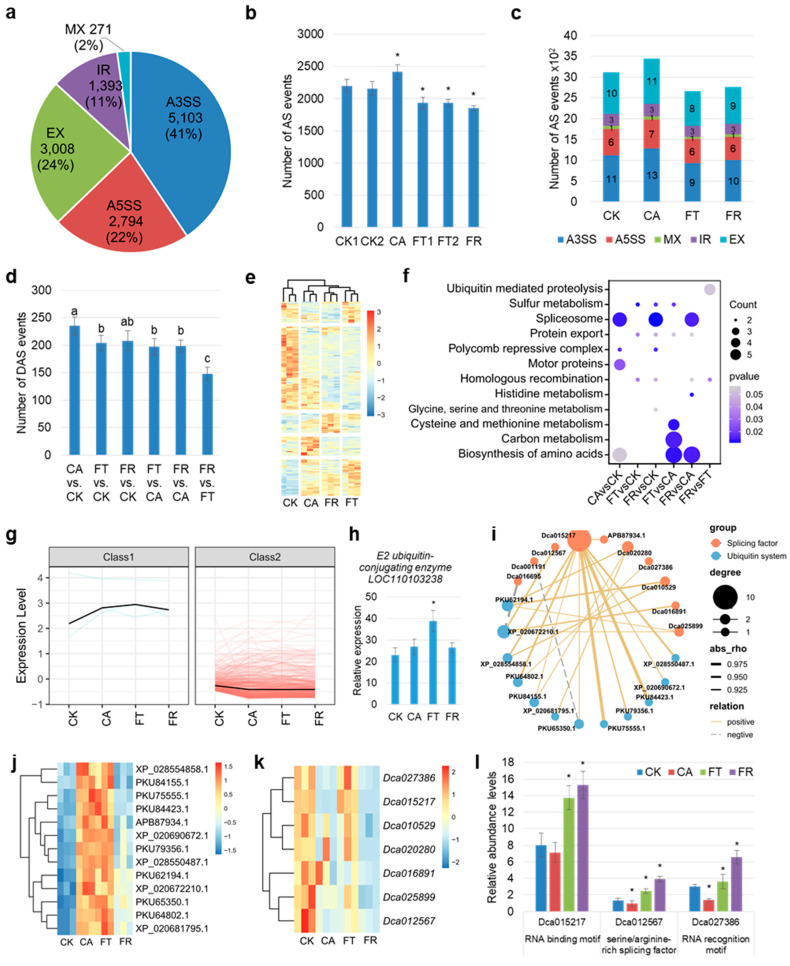
Identification of key splicing factors in response to cold in *D. officinale*. (**a**) The distribution of the five main types of AS events in the *D. officinale* genome (GCF_001605985.2). A3SS (alternative 3′ splice sites), EX (exon skipping), A5SS (alternative 5′ splice sites), IR (intron retention), and MX (mutually exclusive exons) are indicated. (**b**) The number of AS events across different treatments (CK1, CK2, CA, FT, and FR). Cold treatments (FT and FR) showed a significant decrease in the number of AS events compared to the control conditions (CK1/2). Values are means ± S.D. (*n* = 3). Student’s *t*-test, * *p* < 0.05. (**c**) The changes in the number of AS events (A3SS, A5SS, EX, IR, and MX) across treatments. Changes in AS event numbers are indicated by numbers. (**d**) The number of DAS events between different cold treatments and the CK group. Different letters (a–c) indicate significant differences at *p* < 0.05, according to ANOVA analysis and Duncan’s multiple range test. (**e**) Heatmap of the 368 DAS genes exhibiting substantial variation in expression levels across different treatments. The heatmap colors represent the relative expression levels of the genes, with warmer colors indicating higher expression and cooler colors indicating lower expression. (**f**) The functional annotation of the DAS genes, with spliceosome, carbon metabolism, and amino acid biosynthesis pathways being specifically enriched in the CA and FR treatments. (**g**) The expression patterns of the 368 DAS genes grouped into two classes based on their expression levels across different treatments. (**h**) The expression pattern of the DAS gene (*E2 ubiquitin-conjugating enzyme*) that was specifically induced during cold treatment. Values are means ± S.D. (*n* = 3). Student’s *t*-test, * *p* < 0.05. (**i**) Spearman correlation analysis between seven splicing factor genes and 13 ubiquitin proteins. Positive correlations are indicated by yellow lines, while negative correlations are indicated by grey dotted lines. (**j**) Heatmap showing the abundance levels of ubiquitin proteins across different treatments (CA, FT, and FR compared to CK). (**k**) Heatmap showing the expression levels of splicing factor genes across different treatments (CA, FT, and FR compared to CK). (**l**) The abundance levels of three key splicing factor genes encoding RNA binding motif, RNA recognition motif, and serine/arginine-rich protein across different treatments. Values are means ± S.D. (*n* = 3). Student’s *t*-test, * *p* < 0.05.

**Figure 5 plants-14-00412-f005:**
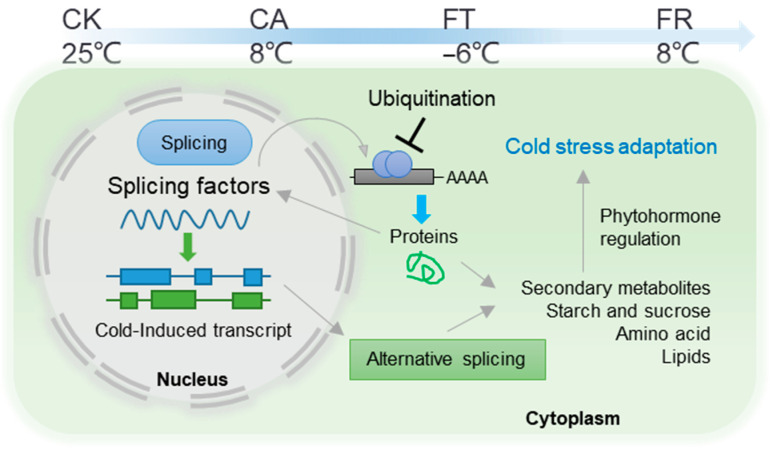
Molecular and metabolic adaptations in *D. officinale* under cold stress. The schematic diagram illustrates the complex interplay between temperature, transcriptomic changes, alternative splicing dynamics, splicing factor regulation, and metabolic adjustments in *D. officinale* under cold stress. It starts with different temperature treatments, including CK (normal temperature conditions), CA (mild cold stress), FT (severe cold stress), and FR (recovery after severe cold stress). These treatments trigger distinct transcriptomic and metabolic responses. Upregulation of spliceosome-related genes and metabolic pathways, such as carbon metabolism and amino acid biosynthesis, occurred primarily in freezing temperature and recovery conditions. In contrast, core metabolic pathways, including nucleotide, amino acid, lipid, and secondary metabolism, are downregulated. There is a rapid increase in AS under mild cold stress, followed by a decrease under severe cold stress. The regulation of splicing factors is depicted through the ubiquitination pathway, which involves ubiquitin proteins and specific splicing factors, such as E2 ubiquitin-conjugating enzyme, zinc knuckle factor, and others like RNA binding motif, RNA recognition motif, and serine/arginine-rich proteins. Metabolic adjustments, including the accumulation of specific metabolites such as oleamide, apigenin glucoside, rutin, and vicenin, are shown to occur in freezing conditions. Arrows and lines represent direct and indirect interactions, respectively, connecting the various components. Colored boxes and text labels provide additional information about each component, illustrating the intricate network of molecular and metabolic adaptations in *D. officinale* under cold stress.

## Data Availability

The original contributions presented in the study are included in the article/Appendix A, further inquiries can be directed to the corresponding author.

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
