# Peer review of "Comprehensive Omics Analysis Reveals Cold-Induced Metabolic Reprogramming and Alternative Splicing in *Dendrobium officinale"

_plants, 2025, doi:10.3390/plants14030412_

Round 1
Reviewer 1 Report
Comments and Suggestions for Authors
Reviewer comments:
Dear Authors,
The submitted manuscript (plants-3438627) is an interesting study exhibiting the importance of metabolic reprogramming and RNA splicing in the cold adaptation mechanism of D. officinale, indicating the activation of a complex network in response to cold stress. The authors carried out a relatively good experiment and obtained satisfactory results. However, the written manuscript still needs further improvement. Here are the main concerns and reviewed comments:
- The quality of English composition in the entire article is not very good and needs improvement. I faced some troubles following the parts due to numerous mistakes in vocabulary.
- The paragraphs of the entire manuscript are written in irregular format. Paragraphs should be modified in consolidated scientific form.
- Gene names should be italicized in the entire text of the manuscript.
- All figures need to be revised in high quality. Font size should be enlarged enough for clear visibility of reading.
- In Abstract, please add more innovative results and their significance for future application.
- The introduction section is too short and lacking the main information about the literature review on previously reported results. Please add supportive literature review.
- In the materials and methods section, a lot of information related to the methodology is missing. Add the detailed information of the adopted methodology, as well as about the characteristics of parental lines used.
- Don’t mention the citation in results; just add them in the introduction, methods, and discussion.
- Add the detail of the PCR reaction protocol of Zhan et al., 2017, or the method used in this study. Also, mention the housekeeping gene.
- The detailed information of synthesized gene information should be uploaded as supplementary material.
- The mapping statistics of RNA-seq data analysis should be uploaded as supplementary tables.
- The discussion part is written in somewhat weak format, and it needs improvement by adding the more comparative and strong description of recently and earlier published results of D. officinale and other comparative crops, explaining the physiological and genetic regulations.
Comments on the Quality of English Language
The English could be improved to more clearly express the research.
Author Response
Dear Authors,
The submitted manuscript (plants-3438627) is an interesting study exhibiting the importance of metabolic reprogramming and RNA splicing in the cold adaptation mechanism of D. officinale, indicating the activation of a complex network in response to cold stress. The authors carried out a relatively good experiment and obtained satisfactory results. However, the written manuscript still needs further improvement. Here are the main concerns and reviewed comments:
- The quality of English composition in the entire article is not very good and needs improvement. I faced some troubles following the parts due to numerous mistakes in vocabulary.
Response: Thank you very much for your comments. The manuscript has been edited by MDPI editing services.
- The paragraphs of the entire manuscript are written in irregular format. Paragraphs should be modified in consolidated scientific form.
Response: Thank you very much for your comments. The manuscript has been edited by MDPI editing services.
- Gene names should be italicized in the entire text of the manuscript.
Response: Thank you very much for your comments. We have revised the content.
- All figures need to be revised in high quality. Font size should be enlarged enough for clear visibility of reading.
Response: Thank you very much for your comments. We have provided the 300dpi figures.
- In Abstract, please add more innovative results and their significance for future application.
Response: Thank you very much for your comments. We have revised the abstract.
- The introduction section is too short and lacking the main information about the literature review on previously reported results. Please add supportive literature review.
Response: Thank you very much for your comments. We have supplemented this section.
- In the materials and methods section, a lot of information related to the methodology is missing. Add the detailed information of the adopted methodology, as well as about the characteristics of parental lines used.
Response: Thank you very much for your comments. We have supplemented the method.
- Don’t mention the citation in results; just add them in the introduction, methods, and discussion.
Response: Thank you for your valuable suggestion. We have revised it.
- Add the detail of the PCR reaction protocol of Zhan et al., 2017, or the method used in this study. Also, mention the housekeeping gene.
Response: Thank you very much for your comments. We have revised the method and provided the sequences in supplementary Table S5.
- The detailed information of synthesized gene information should be uploaded as supplementary material.
Response: Thank you very much for your comments. The sequences were listed in Table S4.
- The mapping statistics of RNA-seq data analysis should be uploaded as supplementary tables.
Response: Thank you very much for your comments. The mapping statistics of RNA-seq were listed in Table S6.
- The discussion part is written in somewhat weak format, and it needs improvement by adding the more comparative and strong description of recently and earlier published results of D. officinale and other comparative crops, explaining the physiological and genetic regulations.
Response: Thank you very much for your insightful comments and suggestions. Although research on cold stress in D. officinale is limited, we have expanded our discussion to include relevant studies on other crops that provide valuable insights into physiological and genetic regulation under cold conditions.

Reviewer 2 Report
Comments and Suggestions for Authors
The manuscript deals with a comprehensive omic analysis of cold induced metabolic reprogramming and alternative splicing in dendrobium officinale. While the general approach suffices there are a few problems with that manuscript that needs to be adressed.
The introduction deals mainly with known effects of cold/decreasing temperatures and the alternative splicing. However the decision to investigate it on Dendrobium is not sufficiently explained. There is no word about the occurence of Dendrobium and its natural habitat, is it usally exposed to cold temperature and could an adaption process already be evolved there? The mentioning of the cultivation is somehow irrelevant since cultivation is normally done under controlled enviroments in greenhouses and cold temperatures don't really play a pivotal part in it. If this is the case and I might be wrong the manuscript introduction failed to make this point more clearly.
A major point that will be also mentioned later is that the authors use abbrevations without introducing them properly. Terms like ROS, MYB, bZIP, WRKY can only be used in a scientific paper after introduction of those, specially in an introduction there is a need to explain what those abbrevations mean.
The results part of the transcriptomic integration is somewhat sound, but also very misleading. The authors write that two control (CK1 and CK2) and two cold treatments (FT1 and FT2) were combined with other cold treatments. What kind of cold treatments were used, if you introduce such things name them, what did you combine??? Were they all used from previous studies?
Are those results all based on previous studies from your own work on D. officinale? Or have new and fresh ones were performed?
The figures look good, only 1c is a bit problematic since the branches all go together and just create a huge black blob, better to remove the branches at some level since the big black mass is more confusing than helping.
The temperature specific response (2.2) part is also sound, but please also don't use WGCNA, KEGG without a proper introduction.
The metabolites part of the paper (2.3) is actually the most problematic. There a few sentence and statement that just don't make any sense "We further identified the high proportion of metabolites in CK, FT and FR" - The reader really has no idea of what you want to say, a higher proportion of metabolites? That sounds like the other small molecules were not metabolites but ghost small molecules. Was it a higher number of metabolites? More metabolites? My imagination is strong and i need it to try and understand this sentence, but i shouldn't need to speculate in a scientific paper about your results. Same thing is for "The kinds of lipids in FT and FR were significantly larger thant those in CK" - do you mean the lipids had longer branches, do you mean the variety of lipids were higher? I have no idea and maybe you consider me a mean reviewer but sentences like this undervalue your whole results since you next sentence starts with describing the differences of FT and FR.
Figure 3 is problematic since it is titled metabolite specific accumulation - but 1g-j deals solely with transcriptomic results of mRNA. Yes these are nice results but are they maybe somehow in the wrong figure?
In the discussion the figure contains basic spelling mistakes "strcach", the reviewers shouldn't be the first to look over a paper and find those kind of mistakes in one of your central figures. While the network is sketched i also can't put the arrow above the figure in a context. The arrow seems very wrong there.
The materials and methods part needs extreme major rehaul. To be honest after reading the materials and methods part i was bordering to reject the manuscript.
The plant material and treatments of Zhan et al 2022 (Metabolites)
"Two-year-old cultivated D. catenatum were grown in soil in the greenhouse of Zhejiang University (Hangzhou, China) under conditions of 25 ± 2 °C (12 h light/12 h dark), 80 μmol photons m−2 s−1, and 65–75% relative humidity [30]. Freezing treatment conditions were performed according to the previous study [24]. In brief, plants were placed at 0 °C and then treated to a gradual drop from 0 to −6 °C within 3 h. The temperature was held at −6 °C for 3 h. Subsequently, the temperature was gradually raised to 8 °C for 12 h and then held at 8 °C for 12 h as a post-freezing recovery treatment. The soil was treated with ice chips to avoid supercooling of the plants. No illumination was provided during freezing treatment and a light of 30 μmol photons m−2 s−1 was provided during the recovery phase. All leaf samples were immediately frozen in liquid nitrogen for proteome and lipidome detection."
It is almost identical to the one here, since it was only more or less copied from your own manuscript this self-plagiarism is not that bad. What is really really bad is that you decided to leave out important parts. Why did you decide to neglect light this time? And "All leaf samples were immeadiatly frozen in liquid nitrogen for proteome and lipidome detection" - That is not only word for word but already very bad in the original. What you want to say is that the leaves were harvested and immeadiatly frozen in liquid nitrogen for analysis. Detection is too vague and could mean millions of things, in terms of reproducability this is not enough. Be precise.
The title about "Multi-omics analysis" is actually the worst part about this. By reading this alone no one has any clue of what you have done, so I immersed myself and thought you must have cited a good paper. So by looking for the transcriptomic procedure i followed the mentioned Zhan et al 2022b, which doesn't mention the procedure but instead cites Zhan et al 2022(another one not in since reference list) and when i followed this rabbit hole and instead of finding the detailed procedure there it cited Than et al 2020 i stoped this process and decided that this is not sound scientific practice. Since the authors have started a more or less spider-like network of only partially mentioning a method and just citing themself for more details which then only partially has parts of the method there i have to insist to break this cycle of never ending self-citations. Please redescribe your full method of how the transcriptomic data was obtained and analyzed from RNAextraction to analysis. Same for lipidome, metabolome and all techniques applied in this paper. Also I am at the moment not conviced that the annotation of the found metabolites and lipids are onehundred percent correct, in one of the rabbit holes you have led me in this paper i found various describtion often mentioning - in house database -, which is problematic in terms of reproducability.
The manuscript needs major improvements, the novelty could be questioned since "Dendrobium Multi-Omics Reveal Lipid Remodeling in Response to Freezing" from the same authors deal with similiar topic and this time the angle was changed. I don't think that this can't be salvaged but since the novelity is already a bit dimished there is no room to overlook the problems mentioned above
Comments on the Quality of English LanguageSee above
Author Response
The manuscript deals with a comprehensive omic analysis of cold induced metabolic reprogramming and alternative splicing in dendrobium officinale. While the general approach suffices there are a few problems with that manuscript that needs to be adressed.
The introduction deals mainly with known effects of cold/decreasing temperatures and the alternative splicing. However the decision to investigate it on Dendrobium is not sufficiently explained. There is no word about the occurence of Dendrobium and its natural habitat, is it usally exposed to cold temperature and could an adaption process already be evolved there? Response: Thank you very much for your comments. We have revised the introduction. The suitable growth temperature for Dendrobium officinale is generally 20 - 25°C, and the safe overwintering temperature is 9 - 12°C. However, wild Dendrobium officinale is widely distributed in southern China. It is still unclear how Dendrobium officinale adapts to the growing environment of different latitudes and different altitudes. Based on our observations, Dendrobium officinale has a strong regenerative capacity after extreme treatment. For instance, its dry stem can be regenerated in a humid environment after being stored at room temperature for a year. The cultivars we studied are located in Zhejiang Province, with winter temperatures droping as low as -10°C sometimes in the wild. Although abiotic stress does not cause widespread Dendrobium officinale death, it seriously affects production. Dendrobium officinale will stop growing after low temperature for a long time.
The mentioning of the cultivation is somehow irrelevant since cultivation is normally done under controlled enviroments in greenhouses and cold temperatures don't really play a pivotal part in it. If this is the case and I might be wrong the manuscript introduction failed to make this point more clearly.
Response: Thank you very much for your comments. Cultivated fields provide only a simple shade, and no greenhouse is used. Greenhouses are too expensive to grow. Thus artificial cultivation of Dendrobium officinale is often threatened by freezing conditions that can damage it in winter.
A major point that will be also mentioned later is that the authors use abbrevations without introducing them properly. Terms like ROS, MYB, bZIP, WRKY can only be used in a scientific paper after introduction of those, specially in an introduction there is a need to explain what those abbrevations mean.
Response: Thank you very much for your comments. We have revised it.
The results part of the transcriptomic integration is somewhat sound, but also very misleading. The authors write that two control (CK1 and CK2) and two cold treatments (FT1 and FT2) were combined with other cold treatments. What kind of cold treatments were used, if you introduce such things name them, what did you combine??? Were they all used from previous studies?
Response: Thank you very much for your insightful comments. We have revised the description of dataset. The cold treatment conditions are descripted in methods. Yes, the transcriptome datasets were from previous studies. We have removed the batch effects of two datasets using the ‘sva’ R package (Leek et al., 2012). The PCA results show that CK and cold treatment are separated into two groups (Figure 1b). Our intention in this study was to reintegrate the transcriptomic data from these two public datasets and explore the core response pathways of Dendrobium officinale under different temperature treatments.
Are those results all based on previous studies from your own work on D. officinale? Or have new and fresh ones were performed?
Response: Thank you very much for your comments. Except for the metabolome, the transcriptome, proteome, and lipidome are from our previous studies. Specifically, one of the transcriptome datasets is from Wu et al., 2016. Previous results were studied separately, without comprehensive analysis. Although some of the omics data were previously published, the comprehensive integration and analysis presented in this manuscript are entirely new. Our study aims to elucidate the core response pathways of cold stress in D. officinale, focusing on key metabolic changes and the regulatory role of the ubiquitination system on AS. We believe these findings significantly advance our understanding of cold adaptation mechanisms in this species.
The figures look good, only 1c is a bit problematic since the branches all go together and just create a huge black blob, better to remove the branches at some level since the big black mass is more confusing than helping.
Response: Thank you very much for your comments. We have revised it.
The temperature specific response (2.2) part is also sound, but please also don't use WGCNA, KEGG without a proper introduction.
Response: Thank you very much for your comments. We have added the annotation.
The metabolites part of the paper (2.3) is actually the most problematic. There a few sentence and statement that just don't make any sense "We further identified the high proportion of metabolites in CK, FT and FR" - The reader really has no idea of what you want to say, a higher proportion of metabolites? That sounds like the other small molecules were not metabolites but ghost small molecules. Was it a higher number of metabolites? More metabolites? My imagination is strong and i need it to try and understand this sentence, but i shouldn't need to speculate in a scientific paper about your results. Same thing is for "The kinds of lipids in FT and FR were significantly larger thant those in CK" - do you mean the lipids had longer branches, do you mean the variety of lipids were higher? I have no idea and maybe you consider me a mean reviewer but sentences like this undervalue your whole results since you next sentence starts with describing the differences of FT and FR.
Response: Thank you very much for your comments. We are sorry for the lack of clarity. We have revised this section.
Figure 3 is problematic since it is titled metabolite specific accumulation - but 1g-j deals solely with transcriptomic results of mRNA. Yes these are nice results but are they maybe somehow in the wrong figure?
Response: Thank you very much for your comments. Given that transcripts are translated into proteins, although the relationship between transcription and translation is not always positively correlated, conducting correlation analysis to explore the relationship between transcripts and proteins is of great significance. It can help us understand how changes at the transcriptional level are ultimately translated into functional changes at the protein level.
In the discussion the figure contains basic spelling mistakes "strcach", the reviewers shouldn't be the first to look over a paper and find those kind of mistakes in one of your central figures. While the network is sketched i also can't put the arrow above the figure in a context. The arrow seems very wrong there.
Response: Thank you very much for pointing out the error. We have revised it.
The materials and methods part needs extreme major rehaul. To be honest after reading the materials and methods part i was bordering to reject the manuscript.
Response: Thank you very much for your comments. We have supplemented and refined the description of the method.
The plant material and treatments of Zhan et al 2022 (Metabolites)
"Two-year-old cultivated D. catenatum were grown in soil in the greenhouse of Zhejiang University (Hangzhou, China) under conditions of 25 ± 2 °C (12 h light/12 h dark), 80 μmol photons m−2 s−1, and 65–75% relative humidity [30]. Freezing treatment conditions were performed according to the previous study [24]. In brief, plants were placed at 0 °C and then treated to a gradual drop from 0 to −6 °C within 3 h. The temperature was held at −6 °C for 3 h. Subsequently, the temperature was gradually raised to 8 °C for 12 h and then held at 8 °C for 12 h as a post-freezing recovery treatment. The soil was treated with ice chips to avoid supercooling of the plants. No illumination was provided during freezing treatment and a light of 30 μmol photons m−2 s−1 was provided during the recovery phase. All leaf samples were immediately frozen in liquid nitrogen for proteome and lipidome detection."
It is almost identical to the one here, since it was only more or less copied from your own manuscript this self-plagiarism is not that bad. What is really really bad is that you decided to leave out important parts. Why did you decide to neglect light this time? And "All leaf samples were immeadiatly frozen in liquid nitrogen for proteome and lipidome detection" - That is not only word for word but already very bad in the original. What you want to say is that the leaves were harvested and immeadiatly frozen in liquid nitrogen for analysis. Detection is too vague and could mean millions of things, in terms of reproducability this is not enough. Be precise.
Response: Thank you very much for your comments. We have supplemented the method. The leaf samples were carefully collected from the treated plants above. To ensure the integrity of the samples and prevent any potential degradation of biological components, all the freshly collected leaf samples were promptly immersed in liquid nitrogen. This immediate freezing step is crucial as it rapidly halts all metabolic activities within the samples, preserving their original molecular states. Subsequently, the samples stored in liquid nitrogen were carefully transferred to the laboratory for omics analysis.
The title about "Multi-omics analysis" is actually the worst part about this. By reading this alone no one has any clue of what you have done, so I immersed myself and thought you must have cited a good paper. So by looking for the transcriptomic procedure i followed the mentioned Zhan et al 2022b, which doesn't mention the procedure but instead cites Zhan et al 2022(another one not in since reference list) and when i followed this rabbit hole and instead of finding the detailed procedure there it cited Than et al 2020 i stoped this process and decided that this is not sound scientific practice. Since the authors have started a more or less spider-like network of only partially mentioning a method and just citing themself for more details which then only partially has parts of the method there i have to insist to break this cycle of never ending self-citations. Please redescribe your full method of how the transcriptomic data was obtained and analyzed from RNAextraction to analysis. Same for lipidome, metabolome and all techniques applied in this paper. Also I am at the moment not conviced that the annotation of the found metabolites and lipids are onehundred percent correct, in one of the rabbit holes you have led me in this paper i found various describtion often mentioning - in house database -, which is problematic in terms of reproducability.
Response: Thank you very much for your detailed and constructive feedback. We understand your concerns regarding the clarity and reproducibility of our methods, particularly in relation to the transcriptomic, lipidomic, and metabolomic analyses. We apologize for any confusion caused by the previous citations and self-references. To address these issues comprehensively, we have redescribed the methods.
The manuscript needs major improvements, the novelty could be questioned since "Dendrobium Multi-Omics Reveal Lipid Remodeling in Response to Freezing" from the same authors deal with similiar topic and this time the angle was changed. I don't think that this can't be salvaged but since the novelity is already a bit dimished there is no room to overlook the problems mentioned above
Response: Thank you very much for your comments. We appreciate your insights and have taken them into careful consideration. To address the concerns regarding novelty and to highlight the unique contributions of our current study, we have revised the manuscript, particularly the abstract, to emphasize new findings and research highlights that distinguish this work from our previous publication.
- Leek, J.T., Johnson, W.E., Parker, H.S., Jaffe, A.E., and Storey, J.D., 2012. The sva package for removing batch effects and other unwanted variation in high-throughput experiments. Bioinformatics 28, 882-883.
- Wu, Z.G., Jiang, W., Chen, S.L., Mantri, N., Tao, Z.M., and Jiang, C.X., 2016. Insights from the Cold Transcriptome and Metabolome of Dendrobium officinale: Global Reprogramming of Metabolic and Gene Regulation Networks during Cold Acclimation. Front Plant Sci 7.

Reviewer 3 Report
Comments and Suggestions for Authors
The study examines how Dendrobium officinale responds to cold stress using transcriptomic, proteomic, and metabolomic analyses. It emphasizes metabolic changes, with cold stress triggering specific metabolites and dynamic alternative splicing, particularly enriched in alternative 3' splice sites (A3SS) and exon skipping (EX). Adjustments in transcriptomics and proteomics reveal differential expression of key genes involved in carbon metabolism, photosynthesis, and secondary metabolite biosynthesis. The study’s objectives are clear, and the results are compelling. While I have no major concerns, I have a few minor suggestions for improvement.
The roles of key factors, such as the E2 ubiquitin-conjugating enzyme and serine/arginine-rich proteins, are suggested but not experimentally validated. Although this aspect is beyond the study's main focus, additional discussion or future plans addressing this limitation would strengthen the manuscript. Similarly, while the correlation analysis between splicing factors and ubiquitination proteins provides valuable insights, causation remains unverified. Including a discussion on potential approaches, such as co-immunoprecipitation or ubiquitination assays, could address this gap. Additionally, the role of specific lipid metabolites in maintaining membrane integrity or signaling under freezing stress is speculative; more discussion on this point would be helpful.
The English writing is generally clear but could be improved for clarity and flow. Simplifying complex sentences and using active voice where appropriate would enhance readability. Transitions between sections, such as from transcriptomics to metabolomics, could be smoother by adding linking sentences to better highlight their connections. Finally, a review by a native English speaker would further refine the manuscript and improve its overall quality.
Lines 19–20: please replace the follow "Key findings revealed that under freezing conditions (FR), specific metabolites such as oleamide, apigenin glucoside, rutin, and vicenin significantly accumulated." to "Under freezing conditions (FR), metabolites such as oleamide, apigenin glucoside, rutin, and vicenin showed significant accumulation."
Lines 20–21: please replace the follow"Lipid species notably increased in both FT (0°C) and FR (-6°C) treatments, with unsaturated glycolipids like monogalactosyldiacylglycerol (MGDG, 36:5) being particularly elevated." to "In both FT (0°C) and FR (-6°C) treatments, lipid species increased significantly, particularly unsaturated glycolipids such as monogalactosyldiacylglycerol (MGDG, 36:5)."
Lines 30–32: please replace the follow "This study underscores the importance of metabolic reprogramming and RNA splicing in the cold adaptation mechanism of D. officinale, highlighting a complex molecular network activated in response to cold stress." to "This study highlights the roles of metabolic reprogramming and RNA splicing in cold adaptation, revealing a complex molecular network activated in response to cold stress."
Lines 37–39: please replace the follow "Low and below-zero environmental temperatures are major environmental challenges that can severely affect crops physiology, leading to reduced growth, altered metabolism, and compromised distribution." to "Low and freezing temperatures pose significant challenges to crops, severely affecting growth, metabolism, and geographical distribution."
Lines 67–68: please replace the follow "Alternative splicing (AS) is a key regulatory mechanism that allows plants to generate diverse transcripts from a single gene, thereby expanding the functional capacity of the transcriptome." to "Alternative splicing (AS) enables plants to produce diverse transcripts from a single gene, significantly enhancing the functional diversity of the transcriptome."
Lines 107–108: please replace the follow "Principal component analysis (PCA) showed that CK partially overlapped with cold treatment (Figure 1a)." to "Principal component analysis (PCA) indicated partial overlap between CK and cold treatments (Figure 1a)."
Lines 112–113: please replace the follow "… showed that 8 core metabolic pathways were down-regulated in cold treatment compared with CK, including nucleotide metabolism, amino acid metabolism, lipid metabolism and secondary metabolism." to "… revealed the downregulation of eight core metabolic pathways during cold treatment, including nucleotide, amino acid, lipid, and secondary metabolism."
Lines 153–154: please replace the follow "The top gene expression levels revealed that D. officinale might have a temperature-specific response by transcriptome integration analysis." to "Transcriptome analysis suggested that D. officinale exhibits temperature-specific gene expression responses."
Lines 348–349: please replace the follow "To bridge the gap in our understanding of the cold stress response in D. officinale, we have carried out an in-depth study of transcriptional regulation and AS regulation in D. officinale under cold stress by using mutil-omics analysis." to "This study addresses gaps in understanding the cold stress response in D. officinale through a multi-omics analysis of transcriptional and AS regulation."
Lines 411–412: please replace the follow "The significant increase in AS events under mild cold stress, followed by a decrease in freezing conditions, suggests a fine-tuned regulatory mechanism that balances splicing patterns to optimize gene expression (Figure 4b)." to "AS events increased during mild cold stress but decreased under freezing conditions, indicating a fine-tuned regulatory mechanism optimizing gene expression (Figure 4b)."
Lines 450–451: please replace the follow "Two-year-old cultivated D. officinale were grown in soil in the greenhouse of Zhejiang University (Hangzhou, China) under controlled conditions in a greenhouse with a 12-hour photoperiod at 25°C/23°C (day/night) and 75% humidity." to "Two-year-old cultivated D. officinale plants were grown in soil under controlled greenhouse conditions (25°C/23°C day/night, 75% humidity, 12-hour photoperiod) at Zhejiang University (Hangzhou, China)."
Comments on the Quality of English LanguageThe English writing is generally clear but could be improved for clarity and flow. Simplifying complex sentences and using active voice where appropriate would enhance readability. Transitions between sections, such as from transcriptomics to metabolomics, could be smoother by adding linking sentences to better highlight their connections. Finally, a review by a native English speaker would further refine the manuscript and improve its overall quality.
Lines 19–20: please replace the follow "Key findings revealed that under freezing conditions (FR), specific metabolites such as oleamide, apigenin glucoside, rutin, and vicenin significantly accumulated." to "Under freezing conditions (FR), metabolites such as oleamide, apigenin glucoside, rutin, and vicenin showed significant accumulation."
Lines 20–21: please replace the follow"Lipid species notably increased in both FT (0°C) and FR (-6°C) treatments, with unsaturated glycolipids like monogalactosyldiacylglycerol (MGDG, 36:5) being particularly elevated." to "In both FT (0°C) and FR (-6°C) treatments, lipid species increased significantly, particularly unsaturated glycolipids such as monogalactosyldiacylglycerol (MGDG, 36:5)."
Lines 30–32: please replace the follow "This study underscores the importance of metabolic reprogramming and RNA splicing in the cold adaptation mechanism of D. officinale, highlighting a complex molecular network activated in response to cold stress." to "This study highlights the roles of metabolic reprogramming and RNA splicing in cold adaptation, revealing a complex molecular network activated in response to cold stress."
Lines 37–39: please replace the follow "Low and below-zero environmental temperatures are major environmental challenges that can severely affect crops physiology, leading to reduced growth, altered metabolism, and compromised distribution." to "Low and freezing temperatures pose significant challenges to crops, severely affecting growth, metabolism, and geographical distribution."
Lines 67–68: please replace the follow "Alternative splicing (AS) is a key regulatory mechanism that allows plants to generate diverse transcripts from a single gene, thereby expanding the functional capacity of the transcriptome." to "Alternative splicing (AS) enables plants to produce diverse transcripts from a single gene, significantly enhancing the functional diversity of the transcriptome."
Lines 107–108: please replace the follow "Principal component analysis (PCA) showed that CK partially overlapped with cold treatment (Figure 1a)." to "Principal component analysis (PCA) indicated partial overlap between CK and cold treatments (Figure 1a)."
Lines 112–113: please replace the follow "… showed that 8 core metabolic pathways were down-regulated in cold treatment compared with CK, including nucleotide metabolism, amino acid metabolism, lipid metabolism and secondary metabolism." to "… revealed the downregulation of eight core metabolic pathways during cold treatment, including nucleotide, amino acid, lipid, and secondary metabolism."
Lines 153–154: please replace the follow "The top gene expression levels revealed that D. officinale might have a temperature-specific response by transcriptome integration analysis." to "Transcriptome analysis suggested that D. officinale exhibits temperature-specific gene expression responses."
Lines 348–349: please replace the follow "To bridge the gap in our understanding of the cold stress response in D. officinale, we have carried out an in-depth study of transcriptional regulation and AS regulation in D. officinale under cold stress by using mutil-omics analysis." to "This study addresses gaps in understanding the cold stress response in D. officinale through a multi-omics analysis of transcriptional and AS regulation."
Lines 411–412: please replace the follow "The significant increase in AS events under mild cold stress, followed by a decrease in freezing conditions, suggests a fine-tuned regulatory mechanism that balances splicing patterns to optimize gene expression (Figure 4b)." to "AS events increased during mild cold stress but decreased under freezing conditions, indicating a fine-tuned regulatory mechanism optimizing gene expression (Figure 4b)."
Lines 450–451: please replace the follow "Two-year-old cultivated D. officinale were grown in soil in the greenhouse of Zhejiang University (Hangzhou, China) under controlled conditions in a greenhouse with a 12-hour photoperiod at 25°C/23°C (day/night) and 75% humidity." to "Two-year-old cultivated D. officinale plants were grown in soil under controlled greenhouse conditions (25°C/23°C day/night, 75% humidity, 12-hour photoperiod) at Zhejiang University (Hangzhou, China)."
Author Response
The study examines how Dendrobium officinale responds to cold stress using transcriptomic, proteomic, and metabolomic analyses. It emphasizes metabolic changes, with cold stress triggering specific metabolites and dynamic alternative splicing, particularly enriched in alternative 3' splice sites (A3SS) and exon skipping (EX). Adjustments in transcriptomics and proteomics reveal differential expression of key genes involved in carbon metabolism, photosynthesis, and secondary metabolite biosynthesis. The study’s objectives are clear, and the results are compelling. While I have no major concerns, I have a few minor suggestions for improvement.
The roles of key factors, such as the E2 ubiquitin-conjugating enzyme and serine/arginine-rich proteins, are suggested but not experimentally validated. Although this aspect is beyond the study's main focus, additional discussion or future plans addressing this limitation would strengthen the manuscript. Similarly, while the correlation analysis between splicing factors and ubiquitination proteins provides valuable insights, causation remains unverified. Including a discussion on potential approaches, such as co-immunoprecipitation or ubiquitination assays, could address this gap. Additionally, the role of specific lipid metabolites in maintaining membrane integrity or signaling under freezing stress is speculative; more discussion on this point would be helpful.
Response: Thank you for your insightful comments and suggestions. We appreciate your feedback, which has prompted us to refine our manuscript further. You rightly pointed out that the roles of key factors such as the E2 ubiquitin-conjugating enzyme and serine/arginine-rich proteins are suggested but not biochemically validated. While this aspect is indeed beyond the immediate scope of our current study due to the challenges associated with Dendrobium officinale not being a model plant, we have already initiated efforts to conduct stable genetic transformations and biochemical experiments. These studies require substantial time and resources, but we are committed to addressing this limitation in future research.
You raised an important point regarding the speculative nature of lipid metabolite` roles in maintaining membrane integrity or signaling under freezing stress. We acknowledge that this is a new discovery that requires further validation. In response, we have revised the discussion section to emphasize the significance of lipid metabolism changes.
The English writing is generally clear but could be improved for clarity and flow. Simplifying complex sentences and using active voice where appropriate would enhance readability. Transitions between sections, such as from transcriptomics to metabolomics, could be smoother by adding linking sentences to better highlight their connections. Finally, a review by a native English speaker would further refine the manuscript and improve its overall quality.
Response: Thank you very much for your comments. The manuscript has been edited by MDPI editing services.
Lines 19–20: please replace the follow "Key findings revealed that under freezing conditions (FR), specific metabolites such as oleamide, apigenin glucoside, rutin, and vicenin significantly accumulated." to "Under freezing conditions (FR), metabolites such as oleamide, apigenin glucoside, rutin, and vicenin showed significant accumulation."
Response: Thank you very much for your comments. We have revised the abstract.
Lines 20–21: please replace the follow"Lipid species notably increased in both FT (0°C) and FR (-6°C) treatments, with unsaturated glycolipids like monogalactosyldiacylglycerol (MGDG, 36:5) being particularly elevated." to "In both FT (0°C) and FR (-6°C) treatments, lipid species increased significantly, particularly unsaturated glycolipids such as monogalactosyldiacylglycerol (MGDG, 36:5)."
Response: Thank you very much for your comments. We have revised the abstract.
Lines 30–32: please replace the follow "This study underscores the importance of metabolic reprogramming and RNA splicing in the cold adaptation mechanism of D. officinale, highlighting a complex molecular network activated in response to cold stress." to "This study highlights the roles of metabolic reprogramming and RNA splicing in cold adaptation, revealing a complex molecular network activated in response to cold stress."
Response: Thank you very much for your comments. We have revised it.
Lines 37–39: please replace the follow "Low and below-zero environmental temperatures are major environmental challenges that can severely affect crops physiology, leading to reduced growth, altered metabolism, and compromised distribution." to "Low and freezing temperatures pose significant challenges to crops, severely affecting growth, metabolism, and geographical distribution."
Response: Thank you very much for your comments. We have revised it.
Lines 67–68: please replace the follow "Alternative splicing (AS) is a key regulatory mechanism that allows plants to generate diverse transcripts from a single gene, thereby expanding the functional capacity of the transcriptome." to "Alternative splicing (AS) enables plants to produce diverse transcripts from a single gene, significantly enhancing the functional diversity of the transcriptome."
Response: Thank you very much for your comments. We have revised it.
Lines 107–108: please replace the follow "Principal component analysis (PCA) showed that CK partially overlapped with cold treatment (Figure 1a)." to "Principal component analysis (PCA) indicated partial overlap between CK and cold treatments (Figure 1a)."
Response: Thank you very much for your comments. We have revised it.
Lines 112–113: please replace the follow "… showed that 8 core metabolic pathways were down-regulated in cold treatment compared with CK, including nucleotide metabolism, amino acid metabolism, lipid metabolism and secondary metabolism." to "… revealed the downregulation of eight core metabolic pathways during cold treatment, including nucleotide, amino acid, lipid, and secondary metabolism."
Response: Thank you very much for your comments. We have revised it.
Lines 153–154: please replace the follow "The top gene expression levels revealed that D. officinale might have a temperature-specific response by transcriptome integration analysis." to "Transcriptome analysis suggested that D. officinale exhibits temperature-specific gene expression responses."
Response: Thank you very much for your comments. We have revised it.
Lines 348–349: please replace the follow "To bridge the gap in our understanding of the cold stress response in D. officinale, we have carried out an in-depth study of transcriptional regulation and AS regulation in D. officinale under cold stress by using mutil-omics analysis." to "This study addresses gaps in understanding the cold stress response in D. officinale through a multi-omics analysis of transcriptional and AS regulation."
Response: Thank you very much for your comments. We have revised it.
Lines 411–412: please replace the follow "The significant increase in AS events under mild cold stress, followed by a decrease in freezing conditions, suggests a fine-tuned regulatory mechanism that balances splicing patterns to optimize gene expression (Figure 4b)." to "AS events increased during mild cold stress but decreased under freezing conditions, indicating a fine-tuned regulatory mechanism optimizing gene expression (Figure 4b)."
Response: Thank you very much for your comments. We have revised it.
Lines 450–451: please replace the follow "Two-year-old cultivated D. officinale were grown in soil in the greenhouse of Zhejiang University (Hangzhou, China) under controlled conditions in a greenhouse with a 12-hour photoperiod at 25°C/23°C (day/night) and 75% humidity." to "Two-year-old cultivated D. officinale plants were grown in soil under controlled greenhouse conditions (25°C/23°C day/night, 75% humidity, 12-hour photoperiod) at Zhejiang University (Hangzhou, China)."
Response: Thank you very much for your comments. We have revised it.

Round 2
Reviewer 1 Report
Comments and Suggestions for Authors
In my opinion, the authors revised the manuscript in satisfactory format. So, the revised manuscript is accepted in its present form.
Thanks
Comments on the Quality of English LanguageThe English could be improved to more clearly express the research.
Author Response
Comments and Suggestions for Authors
In my opinion, the authors revised the manuscript in satisfactory format. So, the revised manuscript is accepted in its present form.
Thanks
Response: We deeply appreciate your meticulous review of our manuscript. Comments on the Quality of English LanguageThe English could be improved to more clearly express the research.
Response: Thank you for your valuable and thoughtful comments. We have carefully checked and improved the English writing in the revised manuscript.
Reviewer 2 Report
Comments and Suggestions for Authors
The authors have sucessfully improved the manuscript, a language review should be carefully performed. The edited version of the revised manuscript sometimes makes it hard to judge the language used.
Author Response
Comments and Suggestions for Authors
The authors have sucessfully improved the manuscript, a language review should be carefully performed. The edited version of the revised manuscript sometimes makes it hard to judge the language used.
Response: Thank you very much for your insightful comments. We deeply appreciate your meticulous review of our manuscript.